# Multiple interaction learning with question-type prior knowledge for constraining answer search space in visual question answering

Tuong Do[1], Binh X. Nguyen[1], Huy Tran[1], Erman Tjiputra[1], Quang D. Tran[1], and Thanh-Toan Do[2]

[1] AIOZ, Singapore
{tuong.khanh-long.do,binh.xuan.nguyen,huy.tran,
erman.tjiputra,quang.tran}@aioz.io
[2] University of Liverpool
thanh-toan.do@liverpool.ac.uk

**Abstract.** Different approaches have been proposed to Visual Question Answering (VQA). However, few works are aware of the behaviors of varying joint modality methods over question type prior knowledge extracted from data in constraining answer search space, of which information gives a reliable cue to reason about answers for questions asked in input images. In this paper, we propose a novel VQA model that utilizes the question-type prior information to improve VQA by leveraging the multiple interactions between different joint modality methods based on their behaviors in answering questions from different types. The solid experiments on two benchmark datasets, i.e., VQA 2.0 and TDIUC, indicate that the proposed method yields the best performance with the most competitive approaches.

**Keywords:** visual question answering, multiple interaction learning.

## 1 Introduction

The task of Visual Question Answering (VQA) is to provide a correct answer to a given question such that the answer is consistent with the visual content of a given image. The VQA research raises a rich set of challenges because it is an intersection of different research fields including computer vision, natural language processing, and reasoning. Thanks to its wide applications, the VQA has attracted great attention in recent years [3, 23, 24, 2, 11, 20]. This also leads to the presence of large scale datasets [3, 7, 10] and evaluation protocols [3, 10].

There are works that consider types of question as the side information which gives a strong cue to reason about the answer [1, 20, 9]. However, the relation between question types and answers from training data have not been investigated yet. Fig. 1 shows the correlation between question types and some answers in the VQA 2.0 dataset [7]. It suggests that a question regarding the quantity should be answered by a number, not a color. The observation indicated that the prior information got from the correlations between question types and answers

open an answer search space constrain for the VQA model. The search space constrain is useful for VQA model to give out final prediction and thus, improve the overall performance. The Fig. 1 is consistent with our observation, e.g., it clearly suggests that a question regarding the quantity should be answered by a number, not a color.

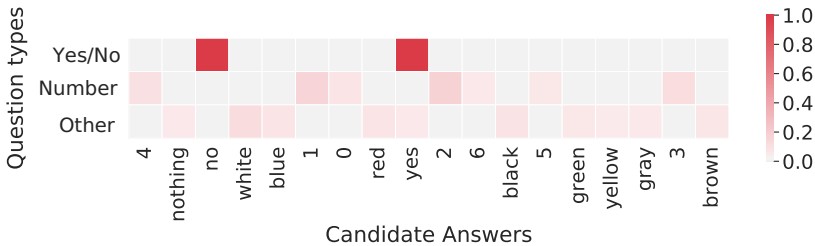

**Fig. 1.** The distribution of candidate answers in each question type in VQA 2.0.

In current state-of-the-art VQA systems, the joint modality component plays an important role since it would learn meaningful joint representations between linguistic and visual inputs [23, 24, 2, 11, 15, 21]. Although different joint modality methods or attention mechanisms have been proposed, we hypothesize that each method may capture different aspects of the input. That means different attentions may provide different answers for questions belonged to different question types. Fig. 2 shows examples in which the attention models (BAN [11] and SAN [24]) attend on different regions of input images when dealing with questions from different types. Unfortunately, most of recent VQA systems are based on single attention models [23, 24, 2, 11, 20, 6]. From the above observation, it is necessary to develop a VQA system which leverages the power of different attention models to deal with questions from different question types.

In this paper, we propose a multiple interaction learning with question-type prior knowledge (MILQT) which extracts the question-type prior knowledge from questions to constrain the answer search space and leverage different behaviors of multiple attentions in dealing with questions from different types.

Our contributions are summarized as follows. (i) We propose a novel VQA model that leverages the question-type information to augment the VQA loss. (ii) We identified that different attentions shows different performance in dealing with questions from different types and then leveraged this characteristic to rise performance through our designed model. (iii) The extensive experiments show that the proposed model yields the best performance with the most competitive approaches in the widely used VQA 2.0 [7] and TDIUC [10] datasets.

## 2    Related Work

**Visual Question Answering**. In recent years, VQA has attracted a large attention from both computer vision and natural language processing commu-

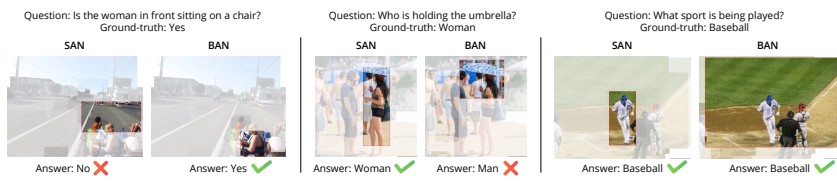

**Fig. 2.** Examples of attention maps of different attention mechanisms. BAN [11] and SAN [24] identify different visual areas when answering questions from different types. ✓ and ✗ indicate correct and wrong answers, respectively.

nities. The recent VQA researches mainly focus on the development of different attention models. In [6], the authors proposed the Multimodal Compact Bilinear (MCB) pooling by projecting the visual and linguistic features to a higher dimensional space and then convolving both vectors efficiently by using element-wise product in Fast Fourier Transform space. In [24], the authors proposed Stacked Attention Networks (SAN) which locate, via multi-step reasoning, image regions that are relevant to the question for answer prediction. In [2, 22], the authors employed the top-down attention that learns an attention weight for each image region by applying non-linear transformations on the combination of image features and linguistic features. In [15], the authors proposed a dense, symmetric attention model that allows each question word attends on image regions and each image region attends on question words. In [11] the authors proposed Bilinear Attention Networks (BAN) that find bilinear attention distributions to utilize given visual-linguistics information seamlessly. Recently, in [21] the authors introduced Cross Modality Encoder Representations (LXMERT) to learn the alignment/ relationships between visual concepts and language semantics.

Regarding the question type, previous works have considered question-type information to improve VQA results. Agrawal et al. [1] trained a separated question-type classifier to classify input questions into two categories, i.e., Yes-No and non Yes-No. Each category will be subsequently processed in different ways. In the other words, the question type information is only used for selecting suitable sub-sequence processing. Shi et al. [20] also trained a question-type classifier to predict the question type. The predicted one-hot question type is only used to weight the importance of different visual features. Kafle et al. [9] also used question type to improve the performance of VQA prediction. Similar to [1], the authors separately trained a classifier to predict the type of the input question. The predicted question type is then used to improve VQA prediction through a Bayesian inference model.

In our work, different from [1], [20] and [9], question types work as the prior knowledge, which constrain answer search space through loss function. Additionally, we can further identify the performance of different joint modality methods over questions from different types. Besides, through the multiple interaction learning, the behaviors of the joint modality methods are utilized on giving out the final answer which further improve VQA performance.

## 3    Methodology

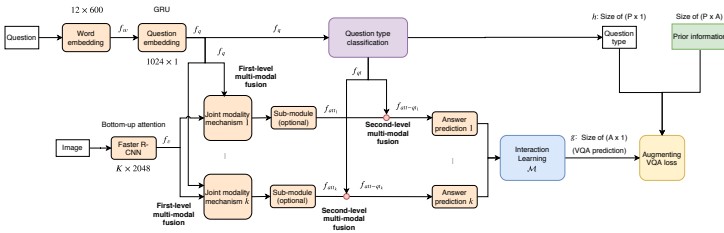

**Fig. 3.** The proposed MILQT for VQA.

The proposed multiple interaction learning with question-type prior knowledge (MILQT) is illustrated in Fig. 3. Similar to the most of the VQA systems [11, 24, 2], multiple interaction learning with question-type prior knowledge (MILQT) consists of the joint learning solution for input questions and images, followed by a multi-class classification over a set of predefined candidate answers. However, MILQT allows to leverage multiple joint modality methods under the guiding of question-types to output better answers.

As in Fig. 3, MILQT consists of two modules: Question-type awareness $\mathcal{A}$, and Multi-hypothesis interaction learning $\mathcal{M}$. The first module aims to learn the question-type representation, which is further used to enhance the joint visual-question embedding features and to constrain answer search space through prior knowledge extracted from data. Based on the question-type information, the second module aims to identify the behaviors of multiple joint learning methods and then justify adjust contributions to giving out final predictions.

In the following, we describe the representation of input questions and images in Section 3.1. Section 3.2 presents the Question-type awareness module $\mathcal{A}$. Section 3.3 presents the Multi-hypothesis interaction learning module $\mathcal{M}$. Section 3.4 presents the multi-task loss for entire model training.

### 3.1    Input Representation

**Question representation.** Given an input question, follow the recent state-of-the-art [2, 11], we trim the question to a maximum of 12 words. The questions that are shorter than 12 words are zero-padded. Each word is then represented by a 600-D vector that is a concatenation of the 300-D GloVe word embedding [17] and the augmenting embedding from training data as [11]. This step results in a sequence of word embeddings with size of $12 \times 600$ and is denoted as $f_w$ in Fig 3. In order to obtain the intent of question, the $f_w$ is passed through a Gated Recurrent Unit (GRU) [4] which results in a 1024-D vector representation $f_q$ for the input question.

**Image representation.** There are several object detectors have been proposed in the literature, of which outputs vary in size and location. Inspired by

recent advances of VQA [2, 20, 22], we use bottom-up attention, i.e. an object detection which takes as FasterRCNN [18] backbone, to extract image representation. At first, the input image is passed through bottom-up networks to get $K \times 2048$ bounding box representation which is denotes as $f_v$ in Fig. 3.

## 3.2  Question-type Awareness

**Question-type classification.** This component in module $\mathcal{A}$ aims to learn the question-type representation. Specifically, aforementioned component takes the question embedding $f_q$ as input, which is then passed through several fully-connected (FC) layers and is ended by a softmax layer which produces a probability distribution $h$ over $P$ question types, where $P$ depends on the dataset, i.e., $P$ equals 3 for VQA 2.0 [7] and equals 12 for TDIUC [10]. The question type embedding $f_{qt}$ extracted from question-type classification component will be combined with the attention features to enhance the joint semantic representation between the input image and question, while the predicted question type will be used to augment the VQA loss.

**Multi-level multi-modal fusion.** Unlike the previous works that perform only one level of fusion between linguistic and visual features that may limit the capacity of these models to learn a good joint semantic space. In our work, a multi-level multi-modal fusion that encourages the model to learn a better joint semantic space is introduced which takes the question-type representation got from question-type classification component as one of inputs.

*First level multi-modal fusion:* The first level fusion is similar to previous works [2, 11, 24]. Given visual features $f_v$, question features $f_q$, and any joint modality mechanism (e.g., bilinear attention [11], stacked attention [24], bottom-up [2] etc.), we combines visual features with question features and learn attention weights to weight for visual and/or linguistic features. Different attention mechanisms have different ways for learning the joint semantic space. The detail of each attention mechanism can be found in the corresponding studies [24, 11, 2]. The output of first level multi-modal fusion is denoted as $f_{att}$ in the Fig. 3.

*Second level multi-modal fusion:* In order to enhance the joint semantic space, the output of the first level multi-modal fusion $f_{att}$ is combined with the question-type feature $f_{qt}$, which is the output of the last FC layer of the "Question-type classification" component. We try two simple but effective operators, i.e. *element-wise multiplication — EWM* or *element-wise addition — EWA*, to combine $f_{att}$ and $f_{qt}$. The output of the second level multi-modal fusion, which is denoted as $f_{att-qt}$ in Fig. 3, can be seen as an attention representation that is aware of the question-type information.

Given an attention mechanism, the $f_{att-qt}$ will be used as the input for a classifier that predicts an answer for the corresponding question. This is shown at the "Answer prediction" boxes in the Fig. 3.

**Augmented VQA loss.** The introduced loss function takes model predicted question types and prior knowledge question types from data to identify the answer search space constraints when the model outputs predicted answers.

*Prior computation.* In order to make the VQA classifier pay more attention on the answers corresponding to the question type of the input question, we use

the statistical information from training data to identify the relation between the question type and the answer. The Alg. 1 presents the calculation of the prior information between the question types and the answers. To calculate the prior, we firstly make statistics of the frequency of different question types in each VQA candidate answer. This results in a matrix $m_{qt-ans}$ (lines 2 to 4). We then column-wise normalize the matrix $m_{qt-ans}$ by dividing elements in a column by the sum of the column (lines 5 to 7).

---

**Algorithm 1:** Question type - answer relational prior computation

**Input** : $Q$: number of questions in training set.
  $P$: number of question types.
  $A$: number of candidate answers.
  $qtLabels \in \{1, ..., P\}^{Q \times 1}$: type labels of questions in training set.
  $ansLabels \in \{1, ..., A\}^{Q \times 1}$: answer labels of questions in training set.
**Output:** $m_{qt-ans} \in \mathbb{R}^{P \times A}$: relational prior of question types and answers.
1  $m_{qt-ans} = zeros(P, A)$ /* init $m_{qt-ans}$ with all zero values */
2  **for** $q = 1 \rightarrow Q$ **do**
3  $\quad | \quad m_{qt-ans}[qtLabels[q], ansLabels[q]] \mathrel{+}= 1$
4  **end**
5  **for** $a = 1 \rightarrow A$ **do**
6  $\quad | \quad m_{qt-ans}[:, a] = normalize(m_{qt-ans}[:, a])$
7  **end**

---

*Augmented VQA loss function design $l_{vqa}$.* Let $y_i \in \mathbb{R}^{A \times 1}$, $g_i \in \mathbb{R}^{A \times 1}$, $h_i \in \mathbb{R}^{P \times 1}$ be the VQA groundtruth answer, VQA answer prediction, and the question-type prediction of the $i^{th}$ input question-image, respectively. Given the question, our target is to increase the chances of possible answers corresponding to the question type of the question. To this end, we first define the weighting (question-type) awareness matrix $m_{awn}$ by combining the predicted question-type $h_i$ and the prior information $m_{qt-ans}$ as follows:

$$m_{awn} = h_i^T m_{qt-ans} \tag{1}$$

This weighting matrix is used to weight the VQA groundtruth $y_i$ and VQA answer prediction $g_i$ to as follows:

$$\hat{y}_i = m_{awn}^T \odot y_i \tag{2}$$

$$\hat{g}_i = m_{awn}^T \odot g_i \tag{3}$$

where $\odot$ is the element-wise product. As a result, this weighting increases the chances of possible answers corresponding to the question type of the question. Finally, the VQA loss $l_{vqa}$ is computed as follows:

$$l_{vqa} = -\frac{1}{QA} \sum_{i=1}^{Q} \sum_{j=1}^{A} \hat{y}_{ij} \log(\sigma(\hat{g}_{ij})) + (1 - \hat{y}_{ij}) \log(1 - \sigma(\hat{g}_{ij})) \tag{4}$$

where $Q$ and $A$ are the number of training questions and candidate answers; $\sigma$ is the element-wise sigmoid function. (4) is a *soft* cross entropy loss and has been shown to be more effective than softmax in VQA problem [22].

It is worth noting that when computing the weighting matrix $a_{awn}$ in (1), instead of using the predicted question type $h_i$, we can also use the groundtruth question type. However, we found that there are some inconsistency between the groundtruth question types and the groundtruth answers. For example, in VQA 2.0 dataset, most of questions started by "how many" are classified with the question type "number", and the answers to these questions are numeric numbers. However, there are also some exceptions. For example, the question *"How many stripes are there on the zebra?"* is annotated with the groundtruth question-type "number" but its annotated groundtruth answer is "many", which is not a numeric number. By using groundtruth question type to augment the loss, the answer to that question is likely a numeric number, which is an incorrect answer compared to the groundtruth answer. In order to make the model robust to these exceptions, we use the predicted question type to augment the VQA loss. Using the predicted question type can be seen as a self-adaptation mechanism that allows the system to adapt to exceptions. In particular, for the above example, the predicted question type may not be necessary "number" and it can be "other".

### 3.3 Multi-hypothesis interaction learning

As presented in Fig. 3, MILQT allows to utilize multiple hypotheses (i.e., joint modality mechanisms). Specifically, we propose a multi-hypothesis interaction learning design $\mathcal{M}$ that takes answer predictions produced by different joint modality mechanisms and interactively learn to combine them. Let $g \in \mathbb{R}^{A \times J}$ be the matrix of predicted probability distributions over $A$ answers from the $J$ joint modality mechanisms. $\mathcal{M}$ outputs the distribution $\rho \in \mathbb{R}^A$, which is calculated from $g$ through Equation (5).

$$\rho = \mathcal{M}\left(g, w_{mil}\right) = \sum_j \left(m_{qt-ans}^T w_{mil} \odot g\right) \tag{5}$$

$w_{mil} \in \mathbb{R}^{P \times J}$ is the learnable weight which control the contributions of $J$ considered joint modality mechanisms on predicting answer based on the guiding of $P$ question types; $\odot$ denotes Hardamard product.

### 3.4 Multi-task loss

In order to train the proposed MILQT, we define a multi-task loss to jointly optimize the question-type classification, the answer prediction of each individual attention mechanism, and the VQA loss (4). Formally, our multi-task loss is defined as follows:

$$l = \alpha_1 \sum_{j=1}^{k} l_{H_j} + \alpha_2 l_{vqa} + \alpha_3 l_{qt} \tag{6}$$

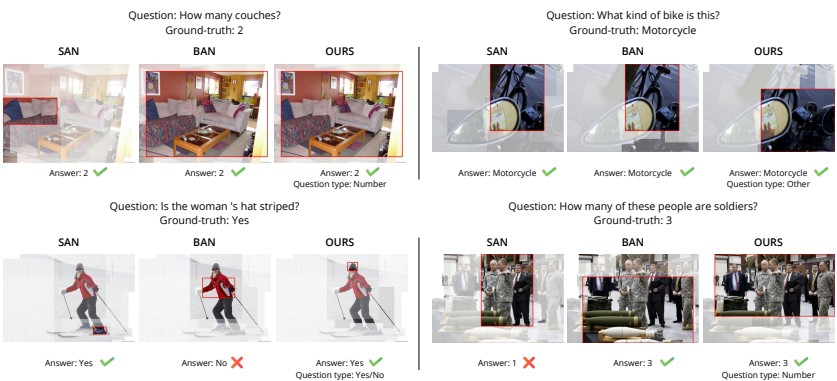

**Fig. 4.** Example results of SAN [24], BAN [11], and our method on the validation set of VQA 2.0. In all cases, the proposed method produces better attention maps. It also produce more accurate answers than compared methods (second row).

where $\alpha_1, \alpha_2, \alpha_3$ are parameters controlling the importance of each loss; $l_{qt}$ is the question-type classification loss; $l_{H_j}$ is the answer prediction loss of $j^{th}$ mechanism over $J$ joint modality methods; $l_{vqa}$ is the introduced VQA loss augmented by the predicted question type and the prior information defined by (4).

## 4  Experiments

### 4.1  Dataset and implementation detail

**Dataset.** We conduct the experiments on two benchmark VQA datasets that are VQA 2.0 [7] and TDIUC [10]. The VQA 2.0 dataset is the most popular and is widely used in VQA problem. In VQA 2.0 dataset, questions are divided into three question types, i.e., "Yes-No", "Number" and "Other" while the TDIUC dataset has 12 different question types.

As standardly done in the literature, we use the standard VQA accuracy metric [3] when evaluating on VQA 2.0 dataset and Arithmetric MPT as well as Harmonic MPT proposed in [10] when evaluating on TDIUC[3].

**Implementation detail.** Our proposed MILQT is implemented using PyTorch [16]. The experiments are conducted on a single NVIDIA Titan V with 12GB RAM.

In all experiments, the learning rate is set to $10^{-3}$ (or $7 \times 10^{-4}$ if using Visual Genome [13] as augmenting data) and batch size is set to 256. The number of detected bounding boxes is set to 50 when extracting visual features. The GRU [4] for question embedding has one layer with 1024-D hidden state and processes words in forward order. During training, except image representations $f_v$, other components are trained end-to-end with the multi-task loss (6). AdaMax optimizer [12] is used to train our model.

---

[3] In [10], the authors show that using Arithmetic MPT and Harmonic MPT is more suitable than the standard VQA accuracy metric [3] when evaluating on TDIUC.

| Models | VQA score |
|---|---|
| **Contribution of question type awareness** | |
| BAN-2-Counter [11] | 65.25 |
| + add | 65.68 |
| + prior | 66.04 |
| + mul | 65.80 |
| + prior | 66.13 |
| **Contribution of hypothesis interaction learning** | |
| BAN-2-Counter [11] | 65.25 |
| + BAN-2 [11] | 66.15 |
| + SAN [24] | 65.64 |
| **Whole model testing** | |
| BAN-2-Counter [11] | 65.25 |
| + BAN-2 [11] + Mul + prior | 66.31 |
| + SAN [24] + Mul + prior | 66.48 |

**Table 1.** Contributions of the proposed components and the whole model on the VQA 2.0 validation set.

| Models | BAN-2 | BAN-2-Counter | Averaging Ens. | Interaction Learning |
|---|---|---|---|---|
| Accuracy | 65.36 | 65.25 | 65.61 | 66.15 |

**Table 2.** Performance on VQA 2.0 validation set where BAN2 [11] and BAN-2-Counter [11] are ensembled using averaging ensembling and the proposed interacting learning.

### 4.2 Ablation study

To evaluate the contribution of question-type awareness $\mathcal{A}$ module and multi-hypothesis interaction learning $\mathcal{M}$ in our method, we conduct ablation studies when training on the train set and testing on the validation set of VQA 2.0 [7].

Starting with the BAN glimpse 2 with counter sub-module (BAN-2-Counter) [11] as the baseline, we show the effectiveness of proposed modules when they are integrated into the baseline. The counter sub-module [25] is used in the baseline to prove the extendability of proposed model on supporting "Number" question. However, any sub-modules can also be applied, e.g., relational reasoning sub-module [19] to support for "Yes/No" and "Other" questions. It is worth noting that in order to make a fair comparison, we use the same visual features and question embedding features for both BAN-2-Counter baseline and our model.

**The effectiveness of question-type awareness and prior information proposed in Section 3.2.** The first section in Table 1 shows that by having second level multi-modal fusion (Section 3.2) which uses element-wise multiplication ($+mul$) to combine the question-type feature $f_{qt}$ and the attention feature $f_{att}$, the overall performance increases from 65.25% (baseline) to 65.80%. By further using the predicted question type and the prior information ($+prior$) to augment the VQA loss, the performance increases to 66.13% which is +0.88% improvement over the baseline. The results in the first section in Table 1 confirm

| Question | Correlation scores | | |
|---|---|---|---|
| types | BAN–Counter | BAN | SAN |
| Yes/No | 0.40 | 0.55 | 0.05 |
| Numbers | 0.55 | 0.23 | 0.22 |
| Others | 0.35 | 0.38 | 0.27 |

**Table 3.** The correlation scores extracted from $w_{mil}$ of MILQT. The extracted information got from model trained in VQA 2.0 train set.

| Models | VQA - test-dev | | | | VQA - test-std | | | |
|---|---|---|---|---|---|---|---|---|
| | Overall | Yes/No | Nums | Other | Overall | Yes/No | Nums | Other |
| SAN [24] | 64.80 | 79.63 | 43.21 | 57.09 | 65.21 | 80.06 | 43.57 | 57.24 |
| Up-Down [2] | 65.32 | 81.82 | 44.21 | 56.05 | 65.67 | 82.20 | 43.90 | 56.26 |
| CMP [21] | 68.7 | 84.91 | 50.15 | 59.11 | 69.23 | 85.48 | 49.53 | 59.6 |
| Pythia [8] | 70.01 | 86.12 | 48.97 | 61.06 | 70.24 | 86.37 | 48.46 | 61.18 |
| BAN [11] | 70.04 | 85.42 | 54.04 | 60.52 | 70.35 | 85.82 | 53.71 | 60.69 |
| LXMERT[21] | **72.4** | 88.3 | 54.2 | 62.9 | **72.5** | 88.0 | 56.7 | 65.2 |
| **MILQT** | 70.62 | 86.47 | 54.24 | 60.79 | 70.93 | 86.80 | 53.79 | 61.03 |

**Table 4.** Comparison to the state of the arts on the test-dev and test-standard of VQA 2.0. For fair comparison, in all setup except LXMERT which uses BERT [5] as question embedding, Glove embedding and GRU are leveraged for question embedding and Bottom-up features are used to extract visual information. CMP, i.e.Cross-Modality with Pooling, is the LXMERT with the aforementioned setup.

that combining question-type features with attention features helps to learn a better joint semantic space, which leads to the performance boost over the baseline. These results also confirm that using the predicted question type and the prior provides a further boost in the performance. We also find out that using EWM provides better accuracy than EWA at the second level fusion.

**The effectiveness of multi-hypothesis interaction learning proposed in Section 3.3.** The second section in Table 1 shows the effectiveness when leveraging different joint modality mechanisms by using multi-hypothesis interaction learning. By using BAN-2-Counter [11] and BAN-2 [11] (BAN-2-Counter + BAN-2), the overall performance is 66.15% which is +0.9% improvement over the BAN-2-Counter baseline.

Table 3 illustrates the correlation between different joint modality mechanisms and question types. This information is extracted from $w_{mil}$ which identify the contributions of each mechanism in giving final VQA results guiding by the question type information.

The results in Table 4 indicate that some joint modality methods achieve better performance in some specific question types, e.g., joint modality method BAN outperform other methods in Number question type by a large margin. The correlation in Table 3 and performance in Table 4 also indicates that the MILQT model tends to leverage the contribution of joint methods proportional to their performance in each specific question type. Besides, the results in Table 2

indicate that under the guiding of question type, $\mathcal{M}$ module produce better performance when comparing with none-use solution or the weighted sum method [14] in which the predictions of different joint modality mechanisms are summed up and the answer with highest score are considered as the final answer.

**The effectiveness of the entire proposed model.** The third section in Table 1 presents results when all components (except the visual feature extractor) are combined in a unified model and are trained end-to-end. To verify the effectiveness of the proposed framework, we conduct two configurations. In the first configuration, we use two joint modality mechanisms BAN-2-Counter and BAN-2, the EWM in the second level multi-modal fusion, and the predicted question type together with the prior information to augment the loss. The second configuration is similar to the first configuration, except that we use BAN-2-Counter and SAN in interaction learning. The third section on Table 1 shows that both configurations give the performance boost over the baseline. The second configuration achieves better performance, i.e., 66.48% accuracy, which outperforms over the baseline BAN-2-Counter +1.23%. Table 1 also show that using "question-type awareness" gives further boost over using interaction learning only, i.e., the performance of "BAN-2-Counter + SAN + Mul + prior" (66.48) outperforms the performance of "BAN-2-Counter + SAN" (65.64). Fig. 4 presents some visualization results of our second configuration and other methods on the VQA 2.0 validation set.

**Question-type classification analysis** The proposed MILQT is a model which allows joint training between question-type classification and VQA answer classification. The effectiveness of multi-task learning helps to improve performance in both tasks. To further analyze the effectiveness of MILQT in the question-type classification, we provide in this section the question type classification on TDIUC dataset. We follow QTA [20] to calculate the accuracy, i.e., the overall accuracy is the number of correct predictions over the number of testing questions, across all categories.

The results are presented in Table 5. Our MILQT uses BAN-2 [11], BAN-2-Counter [11], and SAN [24] in the interaction learning, element-wise multiplication in the second level of multi-modal fusion, and the predicted question type with prior information to augment the VQA loss. Compare to the state-of-the-art QTA [20], our MILQT outperforms QTA for most of question types. In overall, we achieve state-of-the-art performance on question-type classification task on TDIUC dataset with 96.45% accuracy.

It is worth noting that for the "Utility and Affordances" category, the question type classification accuracy is 0% for both QTA and MILQT. It is because the imbalanced data problem in TDIUC dataset. The "Utility and Affordances" category has only $\approx 0.03\%$ samples in the dataset. Hence this category is strongly dominated by other categories when learning the question type classifier. Note that, there are cases in which questions belonging to the "Utility and Affordances" category have similar answers with questions belonging to other categories. Thus, the data becomes less bias w.r.t. answers (in comparing to question categories). This explains why although both MILQT and QTA have 0% accu-

| Question-type accuracy | Reference Models | |
|---|---|---|
| | QTA [20] | MILQT |
| Scene Recognition | 99.40 | **99.84** |
| Sport Recognition | 73.08 | **85.81** |
| Color Attributes | 86.10 | **89.60** |
| Other Attributes | 77.76 | **85.03** |
| Activity Recognition | 13.18 | **16.43** |
| Positional Recognition | 89.52 | **89.55** |
| Sub-Object Recognition | 98.96 | **99.42** |
| Absurd | **95.46** | 95.12 |
| Utility and Affordances | 00.00 | 00.00 |
| Object Presence | **100.00** | **100.00** |
| Counting | 99.90 | **99.99** |
| Sentiment Understanding | 60.51 | **67.82** |
| Overall | 95.66 | **96.45** |

**Table 5.** The comparative question-type classification results between MILQT and state-of-the-art QTA [20] on the TDIUC validation set.

| Score | Reference Models | | | |
|---|---|---|---|---|
| | QTA-M [20] | MCB-A [10] | RAU [10] | MILQT |
| Scene Recognition | 93.74 | 93.06 | 93.96 | **94.74** |
| Sport Recognition | 94.80 | 92.77 | 93.47 | **96.47** |
| Color Attributes | 57.62 | 68.54 | 66.86 | **75.23** |
| Other Attributes | 52.05 | 56.72 | 56.49 | **61.93** |
| Activity Recognition | 53.13 | 52.35 | 51.60 | **65.03** |
| Positional Recognition | 33.90 | 35.40 | 35.26 | **42.31** |
| Sub-Object Recognition | 86.89 | 85.54 | 86.11 | **89.63** |
| Absurd | **98.57** | 84.82 | 96.08 | 88.95 |
| Utility and Affordances | 24.07 | 35.09 | 31.58 | **38.60** |
| Object Presence | 94.57 | 93.64 | 94.38 | **96.21** |
| Counting | 53.59 | 51.01 | 48.43 | **62.41** |
| Sentiment Understanding | 60.06 | **66.25** | 60.09 | 64.98 |
| Arithmetic MPT | 66.92 | 67.90 | 67.81 | **73.04** |
| Harmonic MPT | 55.77 | 60.47 | 59.00 | **66.86** |

**Table 6.** The comparative results between the proposed model and other models on the validation set of TDIUC.

racy for the "Utility and Affordances" on the question category classification, both of them achieve some accuracy on the VQA classification (see Table 5).

### 4.3   Comparison to the state of the art

**Experiments on VQA 2.0 test-dev and test-standard.** We evaluate MILQT on the test-dev and test-standard of VQA 2.0 dataset [7]. To train the model,

similar to previous works [24, 22, 8, 11], we use both training set and validation set of VQA 2.0. We also use the Visual Genome [13] as additional training data.

MILQT consists of three joint modality mechanisms, i.e., BAN-2, BAN-2-Counter, and SAN accompanied with the EWM for the multi-modal fusion, and the predicted question type together with the prior information to augment the VQA loss. Table 4 presents the results of different methods on test-dev and test-std of VQA 2.0. The results show that our MILQT yields the good performance with the most competitive approaches.

**Experiments on TDIUC.** In order to prove the stability of MILQT, we evaluate MILQT on TDIUC dataset [10]. The results in Table 6 show that the proposed model establishes the state-of-the-art results on both evaluation metrics Arithmetic MPT and Harmonic MPT [10]. Specifically, our model significantly outperforms the recent QTA [20], i.e., on the overall, we improve over QTA 6.1% and 11.1% with Arithemic MPT and Harmonic MPT metrics, respectively. It is worth noting that the results of QTA [20] in Table 6, which are cited from [20], are achieved when [20] used the one-hot *predicted question type* of testing question to weight visual features. When using *the groundtruth question type* to weight visual features, [20] reported 69.11% and 60.08% for Arithemic MPT and Harmonic MPT metrics, respectively. Our model also outperforms these performances a large margin, i.e., the improvements are 3.9% and 6.8% for Arithemic MPT and Harmonic MPT metrics, respectively.

We also note that for the question type "Absurd", we get lower performance than QTA [20]. For this question type, the question is irrelevant with the image content. Consequently, this question type does not help to learn a joint meaningful embedding between the input question and image. This explains for our lower performance on this question type.

## 5  Conclusion

We present a multiple interaction learning with question-type prior knowledge for constraining answer search space— MILQT that takes into account the question-type information to improve the VQA performance at different stages. The system also allows to utilize and learn different attentions under a unified model in an interacting manner. The extensive experimental results show that all proposed components improve the VQA performance. We yields the best performance with the most competitive approaches on VQA 2.0 and TDIUC dataset.

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
