# OpenReview forum: "Multiple interaction learning with question-type prior knowledge for constraining answer search space in visual question answering"
_thecvf.com/ECCV/2020/Workshop/VIPriors — VIPriors Poster_

### Official Review · AnonReviewer1 · 2020-07-20
**An ensemble of VQA methods with prior knowledge on question types**

**Confidence:** 3
**Rating:** 7

**Review:**

[Summary] In 2-3 sentences, describe the key ideas, experiments, and their significance.

The authors propose a VQA method that jointly optimizes question answering and answer type classification using an ensemble of existing attention-based VQA methods. Prior information about the answer types is integrated into a weighted loss. Modalities are fused by first merging visual and linguistic features, then merging in question type features.

[Strengths] What are the strengths of the paper? Clearly explain why these aspects of the paper are valuable.

The authors use prior information on answer types while allowing the model to account for outlier questions. The authors perform extensive experiments to show all aspects of their works.

[Weaknesses] What are the weaknesses of the paper? Clearly explain why these aspects of the paper are weak.

It is not clear how significant the individual performance increases of each contribution are. Crucial design choices for the VQA loss are not motivated (e.g. where to integrate awareness matrix). Presentation needs work (typos, grammar, typesetting).

[Overall rating] Paper rating: Accept (tentative rating, subject to revision until deadline)

[Detailed comments] Additional comments regarding the paper (e.g. typos or other possible improvements you would like to see for the camera-ready version of the paper, if any.)

- Please review the paper for typos, incorrect grammar and typesetting. Specifically lines 40 (grammar), 114, 178, 260 "constraints", 264, 287 (capitalizing VQA), 389 "standardly", 436 end of line, 590 "modaality"
- Table 4: your method is not the highest (LXMERT is), so please do not use bold numbers. See also the claim on line 594.
- Algorithm 1 should be expressed in math, like the other equations.
- Source for claim on line 153

---

### Official Review · AnonReviewer2 · 2020-07-27
**Multiple interaction learning with question-type prior knowledge for constraining answer search space in visual question answering**

**Confidence:** 5
**Rating:** 7

**Review:**



#### 1. [Summary] In 2-3 sentences, describe the key ideas, experiments, and their significance.
The paper proposes a method which constrains search space by using question type information as prior information and utilizes different attentions to obtain better results.

#### 2. [Strengths] What are the strengths of the paper? Clearly explain why these aspects of the paper are valuable.
- Search space constraints according to question types
- Using multiple attention mechanisms
- Performance and better attention maps

#### 3. [Weaknesses] What are the weaknesses of the paper? Clearly explain why these aspects of the paper are weak.
- Question types are prior knowledge yet not visual prior knowledge.

#### 4. [Overall rating] Paper rating
7

#### 5. [Justification of rating] Please explain how the strengths and weaknesses aforementioned were weighed in for the rating.


#### 6. [Detailed comments] Additional comments regarding the paper (e.g. typos or other possible improvements you would like to see for the camera-ready version of the paper, if any.)

- Fig.3: Some of the notations are not visible. In addition, you can show modules with dashed areas with different colors.
- Table 4: Why did you make your results as bold?
- The effectiveness of multi-hypothesis interaction learning proposed
in Section 3.3: The explanation in the subsection makes confusion because the order of showing the results. It can be better to have a paragraph for each result (table).

- Limitations and failing cases.
- Time and memory usage
- Will you share the code and models?
Typos:
- L80: constraint
- L.436: not fitting in the line
- L.590:modality

---

### Decision · Program_Chairs · 2020-07-29

**Decision:**

Accept (Poster)

**Comment:**

It is our pleasure to inform you that your paper has been accepted to the poster track of 1st Visual Inductive Priors for Data-Efficient Deep Learning Workshop.

Please note the following deadlines:
* August 11, 2020 - workshop material, including:
 * paper in PDF format;
 * pre-recorded video presentation;
 * slides of the presentation in PDF.
* September 15, 2020 - camera-ready paper

The reviews can be found on OpenReview. Please take these comments and suggestions into account when preparing the camera-ready version of your paper, which is due September 15, 2020. The camera-ready paper should be uploaded to OpenReview.

As part of the workshop, each accepted paper must submit a pre-recorded 90 second talk before August 11, 2020. You will receive more information on how to upload the material shortly. The requirements for the video are:
* Duration: maximum 90 seconds
* MP4 format
* File size max. 100 MB
* Has an inset with a video of the speaker
* 16:9 aspect ratio (strongly preferred)
* 1920x1080 resolution (strongly preferred, at least 720 height)

Our suggested software for pre-recording your presentation is Zoom. For more information, please refer to the following guides:
How to record with Zoom Guide: http://homepages.inf.ed.ac.uk/rbf/ECCV2020HowtoRecordusingZoom.pdf
How to Record with Zoom tutorial: https://www.youtube.com/watch?v=CR199W7HdC0
Please ensure that at least one of the authors of the paper is available to attend the workshop during the allotted times. Note that the workshop will take place in two sessions spread across time zones (details are to follow). We will send instructions on how to connect to the workshop as soon as possible. The schedule for all talks and papers will be posted soon at the workshop website: https://vipriors.github.io.

We look forward to seeing you at the workshop!